# Social world knowledge: Modeling and applications

**Nir Lotan, Einat Minkov**  *

Information Systems Department, University of Haifa, Haifa, Israel

* einatm@is.haifa.ac.il

## Abstract

Social world knowledge is a key ingredient in effective communication and information processing by humans and machines alike. As of today, there exist many knowledge bases that represent factual world knowledge. Yet, there is no resource that is designed to capture *social* aspects of world knowledge. We believe that this work makes an important step towards the formulation and construction of such a resource. We introduce *SocialVec*, a general framework for eliciting low-dimensional entity embeddings from the social contexts in which they occur in social networks. In this framework, *entities* correspond to highly popular accounts which invoke general interest. We assume that entities that individual users tend to co-follow are socially related, and use this definition of social context to learn the entity embeddings. Similar to word embeddings which facilitate tasks that involve text semantics, we expect the learned social entity embeddings to benefit multiple tasks of social flavor. In this work, we elicited the social embeddings of roughly 200$K$ entities from a sample of 1.3M Twitter users and the accounts that they follow. We employ and gauge the resulting embeddings on two tasks of social importance. First, we assess the political bias of news sources in terms of entity similarity in the social embedding space. Second, we predict the personal traits of individual Twitter users based on the social embeddings of entities that they follow. In both cases, we show advantageous or competitive performance using our approach compared with task-specific baselines. We further show that existing entity embedding schemes, which are fact-based, fail to capture social aspects of knowledge. We make the learned social entity embeddings available to the research community to support further exploration of social world knowledge and its applications.

## 1 Introduction

There exist many knowledge bases that describe factual world knowledge. Alongside Wikipedia, which maintains a linked network of textual articles, there are structured knowledge sources like DBPedia [1], Wikidata and Freebase [2], and NELL [3], which describe factual knowledge in relational form. Considering that world knowledge is dynamic and unlimited in scope, researchers have proposed to elicit world knowledge automatically, for example, by inferring entity properties and the relationships between them from the contexts in which they

**Data Availability Statement:** Our source data and code are available in a public data repository located at https://github.com/nirlotan/SocialVecTraining. In addition, the entity embeddings which we generated and experimented with in this work are accessible at https://github.com/nirlotan/SocialVec.

**Funding:** The work received the funding of the Israeli Science Foundation, Grant no. 031/23, awarded to Einat Minkov.

appear in free text [3, 4]. Additional research effort nowadays aims at incorporating the knowledge that is represented in factual knowledge bases into neural information processing and natural language understanding systems, having relevant knowledge about entities and the relations between them encoded in a low-dimensional vector space for this purpose [5–7].

However, the world knowledge that we use in everyday life extends beyond relational facts. Much of the knowledge that is needed for intelligent information processing and communication is in fact *social* in nature. Consider for example the social aspect of political polarity. Knowledge about the political affiliation of persons, organizations and news outlets is necessary for interpreting and identifying possible biases in information that is distributed by them [8]. Likewise, knowledge about other flavors of social affinity or polarity between entities may be needed for interpreting information or conducting a conversation in the respective domains; for example, knowledge about rivalries between sports teams may be needed for conducting a sensible conversation about sports [9]. More generally, it is desired to represent the social and cultural background of the parties involved in communication in order to capture the meaning of information as *intended* and *perceived* by them [10, 11]. And, similar to factual world knowledge, the mapping of social world knowledge is also valuable by its own merit. It can support the organization of entities into broader social structures so as to derive social insights [12], interpret the social aspects involved in contemporary events [13], etc. Yet, we believe that to date, there exist no resource that captures the meaning of entities and the relationships between them from a social perspective. This work makes an important step towards the formulation and construction of such a resource.

Presumably, social network platforms form a valuable source of social world knowledge. To this end, we outline a general approach for eliciting world knowledge from social networks, where we apply and demonstrate this approach using the social network of Twitter. Being a highly popular public social networking service, Twitter has been subject to extensive research across disciplines, and is considered as a credible source of social information [14]. As of today, public figures and organizations, including politicians, artists, and national and local businesses, maintain active presence on social networks in general, and Twitter in particular [15]. Our goal is to learn social representations of such entities from the Twitter network.

Similar to word meaning that is inferred from the neighboring words with which a word commonly co-occurs in text [16], we seek to encode the social meaning of entities based on entity co-occurrences within the social network. We exploit the fact that Twitter users associate themselves with, aka *follow*, other accounts of interest. Naturally, a vast majority of accounts are followed by local social circles, whereas a small minority of the accounts are widely followed. Given a large sample of Twitter users and the accounts that they follow, we identify those accounts that are most popular, considering them as *entities* of general interest. A key observation is that the set of entities that are co-followed by an individual user forms a coherent social context, which reflects the preferences, interests, as well as socio-demographic characteristics of that user. It is well-known, for example, that users typically follow entities of similar political orientation to their own, and seldom follow entities that are affiliated with the opposite camp [17]. Accordingly, we learn entity embeddings by modeling their relatedness with other entities that users tend to co-follow.

We name our approach for learning social entity embeddings as *SocialVec*. Since it models context information at user level, SocialVec enables the learning of entity embeddings from a sample of the social network. In this study, we sampled more than 1.3 million random Twitter users, associating these users with the accounts that they followed. We then learned the embeddings of over 200K entities, which correspond to the top-followed accounts within this data sample, from the sampled network data.

Similar to word embeddings which facilitate tasks that involve text processing, we expect the social entity embeddings to benefit information processing tasks of social flavor. We further expect our framework to support the exploration of entity similarity in the social space, and consequently, to allow researchers and practitioners to derive new insights from this encoding of social knowledge.

A main research question that we address in this paper is therefore, *how to create socially-informative vector representations of a large variety of entities that are of public interest?*

We further explore the following complementary questions: *how do social entity embeddings compare with existing entity embeddings that are derived from factual information sources like Wikipedia?* and, *How can one leverage the learned social entity embeddings in various tasks of interest?*

To address the latter question, we evaluate the social entity embeddings quantitatively–through two different case studies. In the first case study, we formulate and gauge the political bias of news sources in terms of vector similarity within the social embedding space. We compare our results against traditional polls conducted by Pew research [18, 19], showing very high accuracy of our approach. In a second case study, we consider the task of automatically inferring the personal traits of individual users from their social media profiles. In our experiments, we project users onto the social embedding space based on the entities that they follow, and apply supervised learning to predict a variety of personal traits given this user representation, such as *gender*, *ethnicity*, *education level*, and *political leaning*. In both studies, we show competitive performance of our approach compared with viable alternatives. We also compare SocialVec entity embeddings with existing fact-oriented entity embeddings, and show that the latter entity representation schemes lack the social knowledge that is encoded by SocialVec.

We believe that this work forms an important step towards the construction of a general resource of social world knowledge. It presents the following main contributions: (1) We outline SocialVec, a formulation for learning socially-informed low-dimensional entity representations from a sample of the social network. (2) We present empirical details and exploratory results of applying this framework to a large sample of Twitter users, where we identify and learn the representations of 200K entities. (3) The utility of social knowledge modeling is demonstrated in two different case studies. The first study shows that the social orientation of entities, in particular, the political leaning of news sources, can be precisely inferred from the social embedding space by means of vector arithmetic. We then further demonstrate that the social embedding space captures meaningful social context at user level, yielding competitive performance when used as features in personal trait prediction. (4) We show that the knowledge encapsulated in the social embedding space is complementary to existing factual knowledge bases, yielding preferable performance on the explored socially-oriented inference tasks. (5) We make the SocialVec framework and the resulting entity embeddings as learned and applied in this work accessible to the research community, and believe that this has the potential of making a significant contribution to exploring social world knowledge as reflected in Twitter. Our code is available at github.com/nirlotan/SocialVecTraining. Another repository that contains the entity embeddings, as well as an API for assessing entity similarity, is available at https://github.com/nirlotan/SocialVec. (6) The paper outlines future research directions towards automatic social knowledge construction and discusses the potential impact of this resource on related research areas, including applications of Computational Social Science and socially-informed natural language processing.

The paper proceeds as follows. The next section reviews related literature, including a review of existing factual entity embedding schemes. Section 3 outlines our methodology for identifying entities of interest and learning social entity embeddings from social media. In Section 4, we describe the application of our framework to a large sample of Twitter, and explore

the resulting social embeddings space anecdotally. Our results obtained on the tasks of political bias assessment and personal trait prediction are given in Sections 6 and 7, respectively. The paper concludes with a discussion of the implications of this research, future directions, as well as ethical considerations.

## 2 Background and related work

In this section, we first review related approaches for inferring node embeddings in a graph such as the Twitter network, and distinguish our contribution from those works. we then describe existing entity representation schemes, inferred from Wikipedia and Wikidata, which we include in our experiments.

### 2.1 Network embeddings

In their seminal paper, Mikolov *et al.* [16] introduced Word2vec, a neural approach for learning low-dimensional representations of word meaning based on the neighboring words observed in very large amounts of text. Unlike traditional approaches of distributional semantics, Word2Vec is highly scalable, and has been shown to effectively project word semantics onto a compact space of several hundreds of dimensions [20].

The success of Word2Vec has inspired many related works, including in the networks domain. The model of DeepWalk [21] learns latent representation of vertices in a network, using a similar architecture to Word2Vec. Rather than model word co-occurrences within local text sequences, the DeepWalk algorithm samples node sequences from the network via a random walk process, where it is assumed that proximate nodes in the sampled sequences are closely related. The Node2Vec method [22] further generalizes this approach, employing a biased random walk procedure to introduce flexibility in the way a node's neighborhood is defined. Both methods explore the whole graph to learn node embeddings. Accordingly, these methods were applied to relatively small graphs of up to 1M vertices [21, 22]. In this work, we wish to describe popular entities in terms of other entities with which they co-occur on social media. The entities of interest correspond to a small fraction of a very large social network that contains many millions of nodes [23]. Applying transductive algorithms such as DeepWalk would be prohibitively inefficient and practically infeasible for our purposes. Instead, we rely on a sample of users and the entity accounts that they follow. This setting lends itself to a different formulation of node neighborhoods that is efficient and scalable, where we only model relevant co-occurrence statistics among the entities of interest.

Previously, a similar approach was proposed for learning item embeddings from user-item rating history for recommendation purposes [24]. In that work, the embeddings of music items were computed, considering other music items liked by the same users as relevant contexts. Recommendation was then performed based on item similarity in the embedding space. In their experiments, the learned item embeddings were shown to outperform SVD, especially when the rating matrix was sparse.

### 2.2 Entity embeddings

Researchers have previously induced low-dimensional entity representations from factual information sources. Below, we describe in detail two popular and high-performing entity embedding schemes. In our experiments, we will compare our learned social entity embeddings with these entity representations.

**2.2.1 Wikipedia2Vec.** Wikipedia is considered as a valuable resource for learning entity representations due to its scale and the availability of human-curated mapping of entity mentions to their unique identifiers via hyperlinks [5, 25, 26]. The Wikipedia2Vec model learns

word and entity embeddings from Wikipedia, with the aim of placing semantically similar words and entities close to each other in a joint vector space [5, 27]. Concretely, this model applies the Word2Vec formulation to learn representations of word meaning from all of Wikipedia pages. In addition, it incorporates entities into the same semantic space by modeling entity-word and entity-entity interactions. For each hyperlink in Wikipedia, it predicts the words that surround the hyperlink given the referenced entity. And, considering hyperlinked pages as markers of inter-entity relatedness [28], it further predicts the neighboring entities in Wikipedia's link graph. The resulting word and entity embeddings have been successfully applied to a variety of natural language and knowledge processing tasks, including entity linking [29], question answering [30] and knowledge base completion [31], showing preferable performance compared to other relational embedding models [27].

**2.2.2 PyTorch-Big-Graph (PBG): Wikidata graph embeddings.** Wikidata is a popular collaborative knowledge base developed and operated by the Wikimedia Foundation [2]. In addition to entities, Wikidata encodes taxonomic hierarchies ('is a' relationships), entity properties, as well as inter-entity relationships. Overall, it corresponds to a very large graph that includes tens of millions of entities. Lerer *et al* [32] recently presented the distributed PyTorch-BigGraph (PBG) framework, using which they applied multiple graph embedding methods to the very large graph of Wikidata. We will consider their entity embeddings inferred using the popular TransE graph embedding method [33] from the whole Wikidata graph. As reported by Lerer *et al*, the implementation of TransE to Wikidata yielded higher-quality embeddings compared to the DeepWalk algorithm as evaluated on tasks such as link prediction.

We argue that learning entity embeddings from factual sources like Wikipedia and Wikidata fails to model social aspects of world knowledge. In addition, curated knowledge sources like Wikipedia or Wikidata are inherently incomplete [3]. Social networks form a complementary source of world knowledge [34]. We therefore turn to social networks in general, and Twitter in particular, as a large-scale source of social information. To the best of our knowledge, this is the first work that outlines and evaluates an approach for learning entity embeddings with the aim of capturing social world knowledge.

## 2.3 Social inference tasks

In the lack of a common resource of social world knowledge, socially-oriented tasks are addressed ad-hoc, using data and methods that are specific to the problems in question. In this work, we demonstrate how two different tasks can be formulated and processed using our social entity embeddings, showing competitive or preferable results to alternative task-tailored solutions. Below, we introduce these tasks, and the main approaches using which they were addressed in related research.

**2.3.1 Assessing the political bias of news sources.** According to a survey by Pew Research Center, a majority of U.S. adults consume news primarily from social media sites [35]. It is a task of high social importance to infer and communicate the political slant of media sources to users, as the lack of awareness to the underlying political biases may play a role in how news are assimilated and spread on social media [36]. Detection of political biases may further help to address political bubbles, for example, by alerting users of being primarily exposed to ideologically congenial political information [17].

So far, various research works aimed to infer the political slant of media sources based on the language used by them, examining the selection and framing of political issues that are discussed by these sources [37, 38]. Other works quantified the biases of news outlets based on their readership. For example, it has been suggested to analyze the language used by the

followers of each media outlet for assessing its political bias [39]. Otherwise, network-based approaches were employed to assess the ideological similarity between news outlets based on their co-subscribers in Twitter [8]. Ribeiro *et al.* [40] quantified the biases of news outlets using explicit information as provided by Facebook to advertisers about the proportions of liberal and conservative users within the source's audience. In our work, we address this task by exploring the distribution of the news accounts within the learned social embedding space. Specifically, we induce highly accurate assessments of political polarity by assessing the similarity of each news source to accounts of known polarity in the embedding space. Importantly, our proposed approach is generic in that it can be applied to assess political biases of various entities that maintain active social media accounts, as well as other types of social biases.

**2.3.2 Personal trait prediction.** Researchers have long been exploring methods for inferring user profiles automatically from their digital footprints [41]. Such personal information about users is beneficial for applications like personal recommendation [42], as well as for social analytics, where the goal is to infer social insights at scale while considering the user socio-demographic attributes [13, 43]. Existing methods for inferring personal traits typically consider the content associated with the users [44–47]. In particular, multiple works aimed to predict attributes such as *age*, *gender*, *ethnicity*, *education*, *occupation* and *income* from the content posted or consumed by users in Twitter [48–51]. A recent work considered network evidence, modeling users and the accounts that they follow as a bidirectional graph, and applying the DeepWalk algorithm to learn user embeddings in this graph. They then used the resulting embeddings as features in classifying the users' occupation [52]. In a following work, they further introduced the friends of the users into the graph and explored other graph embedding schemes [53]. Both of these works apply semi-supervised transductive learning, inferring the embeddings of labeled and unlabeled nodes jointly. In our work, we project users onto the social knowledge space based on the popular accounts that they follow. This inductive approach is general and scalable, as it does not require ad-hoc sub-network construction or learning of specialized embeddings per user and dataset.

## 3 Methodology

We set the goal of learning social entity embeddings, with the following requirements. First, it is desired to identify popular user accounts that correspond to *entities* of general interest in the realm of social networks. We then wish to map those entities onto a low-dimensional space of social meaning. Importantly, we do not assume there is access or capacity to process information from the whole social network, which is extremely large. That is, data and computation efficiency is a key requirement.

In this section, we formalize our approach to achieving this desiderata. We refer to Twitter as a source of social information, being a popular and public social networking service which has been studied extensively by researchers. Nevertheless, our approach is general, and can be applied to other social networks. Please note that this section concludes with an ethical statement.

### 3.1 Identifying entities of interest from the social network

Users on social networks typically associate themselves with–aka, *follow*–other accounts of interest, to regularly consume the content distributed by them. These associations correspond to a directed graph structure in which vertices denote user accounts and edges represent follower-to-followee relationships. Naturally, a small number of accounts are broadly followed, whereas the vast majority of the users form a long-tail of accounts that are followed by small social circles. We identify the most popular accounts in the social network, as indicated by

their number of followers, considering them to be entities of general interest. To obtain the required statistics, we perform the following steps:

1. We sample a large number of Twitter users $U$ uniformly at random.

2. For each user $u_i \in U$, we obtain the set of accounts that they follow, $\{a_i \mid u_i \rightarrow a_i\}$.

Let $A$ denote the union of all the user accounts that are followed by some user in our sampled sub-network $g$, $A = \cup_i\{a_i\}$. We assess the *popularity* of each account $a_t \in A$ according to the number of users who follow them, $f(a_t) = |\{i \mid a_t \in \{a_i\}\}|$. Finally, we define entities as the subset of the most popular accounts, $E \subset A$, practically setting a threshold $K$ over the number of their followers in $g$, i.e., $a_t \in E$ if $f(a_t) > K$.

## 3.2 Social context modeling

The neural algorithm of Word2Vec builds on the theory of distributional hypothesis in linguistics, by which words that are used in the same contexts tend to have similar meaning [54]. Accordingly, it learns word embeddings that are predictive of neighboring words as observed in large amounts of text. Here, our goal is to learn entity embeddings of social meaning. Similar to local word co-occurrences that demonstrate linguistic and topical regularities, we seek to capture social entity semantics from entity co-occurrences that denote meaningful social contexts.

A key observation that underlies our approach is that the set of entities that are co-followed by an individual user form a coherent unit of social context. Evidently, the links established by a user reflect their personal preferences, interests, and their socio-demographic characteristics. It is well established, for example, that individuals tend to follow news sources with similar political orientation to their own [8, 40]. As we demonstrate later in this paper, the entities that one follows are also correlated with their gender, age group, education level, ethnicity, and more (Sec. 7.3). Thus, similar to word sequences which form linguistically and topically related contexts, the sets of entities followed by individual users demonstrate social and topical inter-relatedness. Accordingly, our objective in learning the entity embeddings would be to predict, for each user and entity pair, the additional entities that are co-followed by the same user. Provided with a large sample of users and the entities that they follow, we expect the neural embedding model to learn meaningful dimensions of social knowledge.

Fig 1 summarizes our approach of data sampling and modeling. As illustrated in the figure, once information about the entities followed by the sampled users is obtained, we discard information about the identities of those users. We further remove information about accounts followed by the sampled users, which are not defined as entities. Notably, entity accounts correspond to a very small portion of the accounts followed, i.e., $E << A$. By that, we diverge from graph embedding approaches like DeepWalk that learn embeddings for all of the nodes in the source graph. Our framework is therefore highly efficient in that it focuses on inter-entity contextual evidence. In Section 4, we detail relevant data statistics observed in a large subgraph sampled from Twitter in practice.

## 3.3 Learning of social entity embeddings

Next, we describe the framework for learning the entity embeddings from the sampled context data. Our approach follows closely on the Word2Vec algorithm. A main difference is that we model large unordered sets of co-occurring entities as context, as opposed to local word neighborhoods that are modeled by Word2Vec.

For clarity, we hereby briefly outline the Word2Vec approach, focusing on the popular skip-gram with negative sampling (SGNS) model [16]. Given unlabeled text corpora,

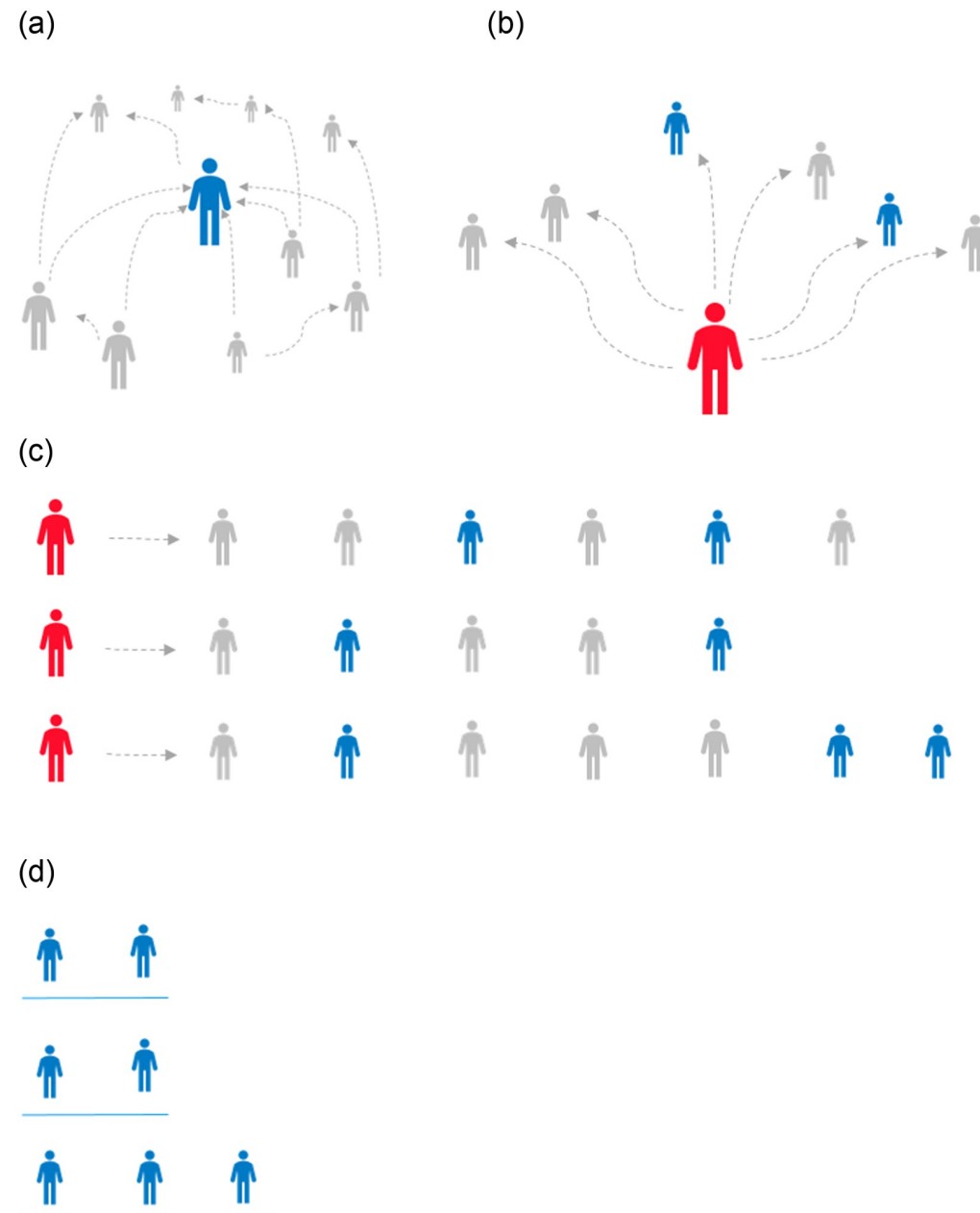

**Fig 1.** A summary of our social context modeling approach: (a) Given a graph of sampled users and the accounts that they follow, we identify accounts of high in-degree, assuming that they represent entities of general interest; the figure illustrates a popular user as a blue figure, and unpopular user–in grey. (b–c) We consider the sets of accounts followed by each sampled user (in red) to be socially related. (d) We focus on entity co-occurrences within these sets, where we discard the sampled users identity, and avoid the modeling of unpopular accounts.

comprised of a sequence of words $(w_i)_{i=1}^{K}$, SGNS learns to predict, for each word $w_i$ in turn, the neighboring words that surround it within a window of a fixed size $c$. The loss function of the model is defined as:

$$L = -\sum_{i=1}^{K} \sum_{-c \leq j \leq c, c \neq 0} log P(w_j \mid w_i) \tag{1}$$

where the conditional probability $P(w_c|w_i)$ is defined using the softmax function:

$$P(w_j \mid w_i) = \frac{exp(u_i^T v_j)}{\sum_{k \in W} exp(u_i^T v_k)} \tag{2}$$

where $u_i \in R^d$ and $v_i \in R^d$ are latent vectors that denote the target and context representations of a given word in the vocabulary, $w_i \in W$. respectively. It is overly costly to compute Eq 2 with respect to the whole vocabulary $W$ however. Negative sampling alleviates this computational burden, replacing the softmax function with:

$$P(w_j \mid w_i) = \sigma(u_i^T v_j) \prod_{k=1}^{N} \sigma(-u_i^T v_k) \tag{3}$$

where $\sigma = (1 + exp(-x))^{-1}$, and $N$ is a parameter. Thus the task becomes to distinguish the target word $w_i$ from a noise distribution, considering $N$ negative samples for each data sample. The negative examples are typically drawn from the unigram word distribution, raised to the 3/4rd power, so as to improve the representation of infrequent words [55]. Finally, training is performed by minimizing the loss function using stochastic gradient descent.

Following our definition of inter-entity relatedness, for each user-entity pair, we with to predict the other entity accounts that are followed by the same user. Accordingly, we define our loss function as follows:

$$L = -\sum_{u_i \in U} \sum_{e_i, e_j \in E_i, e_i \neq e_j} logP(e_j \mid e_i) \tag{4}$$

where $e_i$ and $e_j$ are entity pairs that belong to $E_i$, denoting the set of entities that are followed by user $u_i$. Notably, we attribute equal importance to all of the entities that co-occur within the set $E_i$, modeling the correspondences between all of the respective entity pairs. By that, we take full advantage of our reference data. Alternatively, one may opt for a stochastic approach, modeling relatedness between sampled entity pairs [24].

In addition to the skip-gram neural model, we consider the CBOW model variant of Word2Vec [16]. Using CBOW, the goal is to predict each focus word $w_i$ in turn given the aggregate representation (sum of embeddings) of its neighboring words. Similarly, we aim to each focus entity $e_i$ from the aggregate representation of all of the other entities that are followed by the same user. Due to the aggregation operation, CBOW is substantially faster to train compared with SGNS. Performance-wise, SGNS has been found to give comparable results on syntactic tasks, and preferable results to CBOW on tasks that model semantic similarity between words [16, 56]. We considered both variants and compare between them in our experiments.

## 3.4 Ethics statement

The proposed framework relies on public social network information. As described in Sec. 4, we obtained relevant network information for sampled users via a public Twitter API under the terms of Twitter developer account. Our automated data processing framework discards user identities (Sec. 3.2), and projects the large-scale anonymized user data onto a low-dimensional space (Sec. 3.3). For these reasons, personal information cannot be recovered from the generated entity embeddings, which rather reflect global social patterns. The data which we obtained for the purpose of this research may be recovered by other researchers using the same procedure and under the same terms. Details about our user sample, as well as our code

used for extracting relevant network information and learning the entity embeddings, are available at github.com/nirlotan/SocialVecTraining.

## 4 SocialVec: Application and evaluation

It is generally desired to train neural models using abundant data that is of high-quality and representative of the target data distribution. We applied our framework to a large sample of Twitter network information, comprised of over a million users and the accounts that they follow. We sampled the user identifiers uniformly at random from a pool of users in the U.S. who posted tweets in the English language. (Concretely, we sampled the users from a corpus of tweets authored by over 10 million users in 2015 that was acquired from Twitter for research purposes.) We then retrieved the full list of accounts followed by each user using Twitter API. (We used the tweepy wrapper library [57] for this purpose.) This network data was collected in the beginning of 2020. Overall, our sample includes 1.3 million distinct users and 1,236 million relationships, mapping to 90.4 million unique accounts that are followed by the users in the sample.

In this section, we describe the implementation details and results of applying each part of our framework to this data, including entity identification, context extraction, and the learning of the entity embeddings. Supplementary information, including our code and the list of user identifiers that comprise our sample of Twitter users, are included in a data depository. Researchers may re-obtain similar data from Twitter, reproduce relevant embeddings or perform related research using this resource.

This section concludes with an exploratory examination of the learned social embedding space. Quantitative evaluation, in which the learned entity embeddings are used to perform social inference tasks of interest, are included in the following sections.

### 4.1 Account popularity statistics

First, let us examine the distribution of accounts by their popularity, and discuss the procedure of defining popular accounts as entities. Fig 2 shows the distribution of account popularity as measured by the number of users in our sample who follow each account. As shown, a small number of accounts (1.4K) are followed by more than 25,000 users each, i.e., by $\sim 2\%$ of the users in our dataset. Roughly $\sim 5.5$K accounts are followed by 10K users or more, i.e., by more than 0.8% of the users. As expected, the vast majority of accounts form a long tail of the popularity distribution, being followed by less than 1K users. Learning the embeddings of all the accounts would be challenging computationally as well as quality-wise, considering that sufficient contextual information is needed for learning meaningful semantic representations. Furthermore, we wish to focus on widely-known accounts for our purpose of social knowledge modeling. We therefore set a threshold over account popularity, considering the accounts that are followed by at least $k = 350$ users ($\sim 0.03\%$ of the users in our sample) as entities. There are roughly 200K accounts (201,247) which meet this condition and comprise our vocabulary of entities, $E$.

A question of interest is, *to what extent does our Twitter-based vocabulary of entities E represent entities that are indeed of general interest?* To answer this question, we assess the extent to which the identified entities $E$ are included in existing sources of world knowledge. Admittedly, knowledge bases are incomplete in their coverage. Nevertheless, the inclusion of an entity by other sources of world knowledge may be considered as an indication of public interest in that entity.

Exploiting the fact that the collaboratively-managed Wikidata links entities with their respective page on Wikipedia as well as with their account handle in Twitter, we aligned the

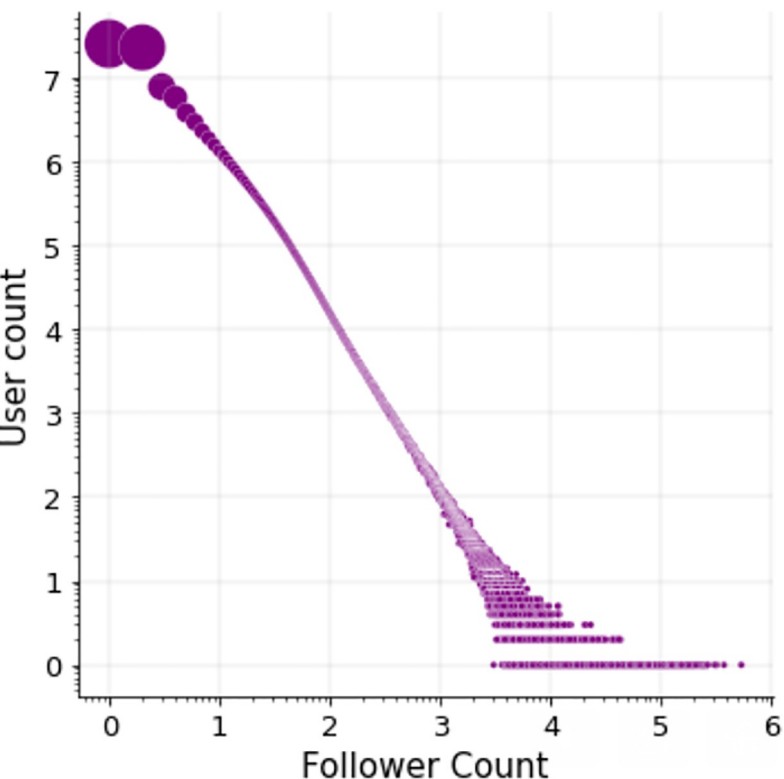

**Fig 2. Account popularity based on our sample of 1.3 million Twitter users and the accounts that they follow: Number of accounts vs. number of their followers (log-log scale, where the number of accounts is illustrated by point size).** There are 90 million unique accounts with at least one follower in the sampled data. A long tail of accounts are followed by up to 100 users (top-left part of the plot). We learn the embeddings of the most popular ~200K accounts, which have 350 followers or more.

entities $E$ whenever applicable with Wikidata and Wikipedia. (See Wikidata property: P2002, and twitter user numeric id property P6552. We used Wikidata's SPARQL query service available at https://pypi.org/project/qwikidata/ to retrieve this information.) Overall, we found that 31.5K out of the 200K accounts that comprise $E$ (15.8%) have a Wikipedia page or Wikidata entry associated with them. Naturally, there is a correlation between account popularity on Twitter and knowledge base coverage. Among the 10K most popular accounts in $E$, more than 50% are included in Wikidata, where this ratio goes down to 5% for the 10K entities that are least popular. Considering that the coverage of the reference knowledge bases and the alignment to these sources are imperfect, these figures form a conservative and encouraging assessment of entity relevance. Notably, there are differences between formal knowledge bases and social networks like Twitter with respect to representation scope. In manual examination, we observed that some popular accounts in Twitter concern professional topics or hobbies; these accounts carry social information but are of low relevance to factual knowledge bases. On the other hand, knowledge bases like Wikipedia include many abstract, scientific, and historical concepts, which are not present in social networks.

## 4.2 Context statistics

Table 1 includes statistics about the number of accounts followed by individual users in our dataset. As shown, users follow almost 1,000 other accounts on average, where the median

**Table 1. Statistics regarding the number of accounts followed by individual users in our sampled network, considering: All accounts, popular accounts which we consider as entities, and entity accounts for which a respective entry in Wikipedia has been found.**

|  | All | Entities | Wikipedia-aligned |
|---|---|---|---|
| Average | 978 | 228 | 116 |
| Std.dev | (1373) | (335) | (175) |
| Median | 526 | 108 | 56 |

number of accounts that a user follows is 526. As shown in the table, out of those, users follow 228 popular accounts that included in our entity vocabulary $E$ on average, where the median number of entities that a user follows is 108. As further shown, more than half of the entities followed by the users have a respective entry in Wikipedia. This means that our approach models as context entities that are well-known alongside entities that are mainly popular on social media.

## 4.3 Learning setup

To learn the entity embeddings, we adapted the Word2vec algorithm as implemented in gensim [58]. Importantly, we set the context size parameter to a large value, $c = 1000$. Since most users follow a substantially smaller number of entities (228 on average; see Table 1), this means that the model is practically trained to predict all of the pairwise entity co-occurrences within the context of every user (Eq 4). We applied common parameter choices in training the model, setting the initial learning rate to 0.03, and having it decrease gradually to a minimum value of 0.0007. Considering the large context size on one hand, and computational requirements on the other hand, we increased the number of negative examples to $N = 20$, which we selected randomly from the probability distribution of the entities followed in our data, applying downsampling by a factor of 1e-5. Similar to Wikipedia2Vec entity embeddings, we set the size of the social entity embeddings to 100 dimensions. In our experiments, we created and evaluated embeddings of different sizes based on cross-validation performance on the task of personal trait prediction (Section 7), and found that 100-dimensions provide comparable performance to higher-dimension representations.

The training of the models was conducted using an Intel(R) Core(TM) i9-7920X CPU @ 2.90GHz computer with 24 CPUs, with 128GB RAM and an NVIDIA Corporation GV100 [TITAN V] (rev a1) GPU. Generating the entity embeddings using the SGNS model required about five days in running time. In addition, we trained social embeddings using the CBOW configuration, which was about five times slower.

## 5 Evaluation

### 5.1 Evaluation methodology

**5.1.1 Intrinsic vs. extrinsic evaluation.** Word embedding schemes are often evaluated against human-labeled benchmarks, where word pairs that are judged to be highly similar by humans are expected to demonstrated high similarity in the embedding space, and vice versa [59]. Unfortunately, there are no relevant human-judged benchmarks that assess entity similarity. Instead, we take a direct look at inter-entity similarities by exploring the entities that are most similar to example entities of interest in the learned social embedding space. We believe that this exploratory study highlights several types of social information that are learned using our approach. This intrinsic evaluation is presented and discussed in Section 5.2.

Nevertheless, we mainly place our focus on extrinsic evaluation, where we gauge the utility of the learned social entity embeddings for end applications which involve social inference. First, we assess the political bias of news sources in terms of entity similarity in the social embedding space. Second, we predict the personal traits of individual Twitter users based on the social embeddings of entities that they follow. These studies are presented in Sections 6 and 7, respectively.

**5.1.2 Comparison against alternative methods and baselines.** In evaluating SocialVec on end tasks, we review and compare our results against alternative methods, which have been designed and applied per those target tasks and experimental datasets. In addition, we contrast our social entity embeddings with existing entity embedding schemes, namely, Wikipedia2Vec and Wikidata RBG embeddings. For our experiments, we obtained the 100-dimension Wikipedia2Vec entity embeddings based on a dump of Wikipedia in English from October 2020 using the code provided by Yamada *et al*. [27]. In addition, we experiment with distributed the 200-dimension RGB entity embeddings as pre-trained using the TransE method and a dump of Wikidata from 2019-03-06 [32]. To the extent that SocialVec achieves higher performance on tasks of social inference, this implies that it is superior in capturing dimensions of social meaning.

## 5.2 Exploratory evaluation of entity similarity

To illustrate the learned social semantics, we consider several example entities, exploring other entities in their vicinity. Table 2 lists the top (5) entities that were found to be closest in terms of cosine similarity to the entities of *Pfizer*, *Princeton*, *X-Men*, *Star Trek* and *Hillary Clinton*. In addition to SocialVec, the table also details the most similar entities to each query entity using Wikipedia2Vec and Wikidata RBG embeddings.

We observe that SocialVec models semantic similarity. For example, the closest entities to *Pfizer* include other pharmaceutical and Biotechnology companies, including *Amgen, Novartis, Sanofi*, and *AstraZeneca*. The account of 'Pharmaguy' is topically-related–it is a newsletter on pharmaceutical marketing. Likewise, the entities that are most similar to *Princeton University* include other prestigious universities, most of which are also members of the Ivy league. And, the most similar entities to *X-Men* are other movies that depict fictional superheroes by Marvel. The most related entities to *Star Trek*, which is defined by Wikipedia as a pop-culture franchise, include actors in the Star Trek movie series. (Dan Aykord is rather known for playing a role in a satire sketch of Star Trek.) These results suggest that modeling user preferences on social media is useful for eliciting social and cultural, as well as semantic inter-entity similarity. Lastly, considering *Hillary Clinton* as the focus entity well-demonstrates the encoding of political social knowledge within SocialVec. The most related entities in this case include the Democratic former first lady *Michelle Obama*, as well as the Democratic Senators *Elizabeth Warren* and *Cory Brooker*. In addition, *Clinton* is found similar in the embedding space to the *Planned Parenthood organization* and *Jim Acosta*, a CNN journalist. These accounts, which span over politicians, organizations, and the media, all belong to the Democratic political camp.

Considering the top entities retrieved using Wikipedia2Vec, we observe somewhat different semantics. Similarly to SocialVec, Wikipedia2Vec places *Pfizer* next to pharmaceutical and Biotechnology companies, and *X-Men*–next to other superheroes movies by Marvel. But, the most related entities to 'Star Trek' include movies and some fictional characters (e.g., 'James Kirk') that are not well-represented on Twitter. Further, the closest entities to *Princeton University* according to Wikipedia2Vec include the city of Princeton, as well as the *Institute for Advanced Studies* and *Nassau Hall* building. Indeed, these entities are affiliated with Princeton University, and are directly linked with the university page on Wikipedia. Yet, we find that

**Table 2. The most similar entities to exemplary query entities, as ranked using cosine similarity by the different entity embedding methods.**

| | SocialVec | wikipedia2Vec | Wikidata:TransE |
|---|---|---|---|
| **Pfizer** | | | |
| 1 | Amgen | Bristol Myers Squibb | Merck & Co. |
| 2 | Novartis | Merck & Co. | Johnson & Johnson |
| 3 | pharmaguy | GlaxoSmithKline | American International Group |
| 4 | Sanofi US | Genzyme | Cisco Systems |
| 5 | AstraZeneca | Eli Lilly and Company | Bank of America |
| **Princeton University** | | | |
| 1 | Columbia University | Princeton School of Public and International Affairs | Stevens Institute of Technology |
| 2 | Yale University | Institute for Advanced Study | Swarthmore College |
| 3 | Cornell University | List of Princeton University people | Brandeis University |
| 4 | Brown University | Princeton, New Jersey | Yale Law School |
| 5 | Cambridge University | Nassau Hall | Woodrow Wilson |
| **X-Men** | | | |
| 1 | Iron Man | Professor X | X-Men 2 |
| 2 | Captain America | X-Force | X-Men: The Last Stand |
| 3 | Guardians Of The Galaxy | Avengers | Real Steel |
| 4 | Thor | Magneto (Marvel) | X-Men: First Class |
| 5 | The Avengers | Age of Apocalypse | Seven Brides for Seven Brothers *(musical film)* |
| **Star Trek** | | | |
| 1 | Jeri Ryan | Star Trek: The Original Series | Star Trek: Nemesis |
| 2 | Gates McFadden | Star Trek: The Next Generation | Star Trek: Insurrection |
| 3 | Michael Dorn | Star Trek: Deep Space Nine | Star Trek: First Contact |
| 4 | Jonathan Frakes | Star Trek: Voyager | NCAA Basketball Tournament Most Outstanding Player |
| 5 | Dan Aykroyd | James T. Kirk | Star Trek: Enterprise |
| **Hillary Clinton** | | | |
| 1 | Senator Elizabeth Warren | 2016 United States presidential election | Barack Obama |
| 2 | Michelle Obama | Barack Obama | John Quincy Adams |
| 3 | Planned Parenthood Federation of America (PPFA) | Joe Biden | John Kerry |
| 4 | Jim Acosta | Donald Trump | Bill Clinton |
| 5 | Senator Cory Booker | John Kerry | United States Secretary of State |

these responses fail to capture the social perception of Princeton as a prestigious university. For *Hillary Clinton*, Wikipedia2Vec assigns top similarity to other Presidential candidates, as well as the page of '2016 United States presidential election'. This notion of similarity fails to capture political affinity, notably placing the Republican Trump in close vicinity to Clinton. In the case of Wikidata RBG graph embeddings, we find some bias towards functional similarity. For example, *Cisco* and *Bank of America* are placed close to *Pfizer*, despite being companies in different sectors. Or, *President Adams* is highly similar to *Hillary Clinton* who ran for Presidency, despite being a historical figure. Overall, we find that unlike these existing entity representation schemes, SocialVec reflects popular social knowledge and conception of entities.

## 6 Case study I: Political polarity of news sources

We introduced the task of inferring the political slant of media sources in Section 2.3. In this case study, we investigate whether the political orientation of news sources is encoded in the

inferred social embeddings space, where we frame and assess political bias in terms of social entity similarity. We validate our predictions against two large polls by Pew Research, showing high correlation with their findings.

## 6.1 Method

Let us recall that our entity embeddings were learned from context information comprised of popular accounts that are co-followed by users on social media. Presumably, individual users follow the accounts of media sources, politicians, and other entities with similar political orientation to their own. It is therefore expected that entities that are associated with the same political camp demonstrate high similarity in the embedding space, as opposed to entities of opposing camps. Accordingly, we compute the bias of news accounts based on their similarity to popular anchor accounts that represent the two political poles. Specifically, we consider the accounts of Barack Obama, the former Democratic U.S. president and the incumbent president at the time that our data was collected, the Republican Donald Trump. As of 2020, these accounts were ranked as first and fourth most popular Twitter accounts based on the number of their followers. (Trump's account was suspended in January 2021).

Let us denote the embedding of a specified news account as $e_n$, and the embeddings of the Democratic and Republican anchor accounts, which we set to the accounts of Obama and Trump, as $e_D$ and $e_R$, respectively. We first measure the similarity of the news source in the embedding space with respect to these Republican and Democratic anchors, and then assess the *political orientation (PO)* of account $e_n$ as the difference between the two similarity scores:

$$PO(e_n) = Sim(e_R, e_n) - Sim(e_D, e_n) \tag{5}$$

A positive score indicates on overall conservative (Republican) orientation, and a negative score–on a liberal (Democratic) bias. A greater gap between the similarity scores suggests there is a greater political bias of entity $e_n$.

In our experiments, we rank specified news accounts according to their computed political orientation scores, assessing our results against formal polls. We experiment with SocialVec entity embeddings, as well as with the alternative embeddings schemes. The success of each entity representation method on this task reflects the extent to which it captures the social phenomena of political leaning.

## 6.2 Ground-truth data

Two polls were conducted by Pew Research in 2014 and 2019 with the goal of gauging the political polarization in the American public [19]. The participants in the polls were recruited using random sampling of residential addresses, and the data was weighted to match the U.S. adult population by gender, race, ethnicity, education and other categories. In both polls, Pew researchers classified the audience of selected popular news media outlets based on a ten question survey covering a range of issues like homosexuality, immigration, economic policy, and the role of government. The media sources were then ranked based on the party identification (Republican or Democrat) and ideology (conservative, moderate or liberal) of the survey participants.

Overall, the polls conducted in 2014 and 2019 include 36 and 30 news media outlets, respectively. The union of these two sets comprises 43 unique media outlets. We were able to identify the Twitter handles of most of the news sources included in the earlier poll (31/36), and practically all of the news sources included in the more recent poll (30/30). As detailed in Table 3, all of these media accounts are encoded as entities in SocialVec, where most albeit not all of the accounts are represented by Wikipedia2Vec and Wikidata RBG embeddings.

**Table 3. Results of assessing the political bias of news sources.** The table reports Spearman's correlation of conservative-to-liberal ranking of selected news accounts generated based on different entity embeddings, compared with poll-based rankings reported by Pew Research in 2014 and 2020. The number of relevant account embeddings is given in parenthesis for each method.

|  | Twitter accounts | SocialVec | Wikipedia2Vec | Wikidata RBG |
|---|---|---|---|---|
| Pew Research poll, 2014 | 31 | 0.82 (31) | 0.36 (28) | -0.40 (23) |
| Pew Research poll, 2020 | 30 | 0.85 (30) | 0.28 (28) | -0.32 (23) |

## 6.3 Results

**6.3.1 Ranking-based evaluation.** The surveys by Pew Research assign a numerical score to each of the news sources that reflects their political polarity. In assessing our method, we therefore consider the relative ranking of the various news sources–ranging from conservative/Republican to liberal/Democrat. Table 3 reports the alignment between the poll-based rankings and the rankings generated according to Eq 5 using different entity embedding methods, measured in terms of Spearman's correlation [59]. A perfect Spearman correlation of +1 indicates that the rankings are identical, whereas a correlation of -1 means that they are perfectly inverse. As shown in the table, the rankings produced using SocialVec closely match the ground-truth rankings, yielding high Spearman's correlation scores of 0.82 and 0.85 per the two polls. In contrast, the rankings produced by Wikipedia2Vec yield low scores of 0.36 and 0.28. The rankings generated using graph-based Wikidata embeddings are not meaningful altogether, yielding negative correlation scores.

Fig 3 illustrates the distribution of political orientation scores of news sources included in the poll of 2020 as computed using the SocialVec embeddings. The accounts are spaced along the range of Democratic (left) to Republican (right) according to their scores. Similar to the reference poll results, we observe that some news sources lie very close to each other on this scale of political bias. This means that using the measure of Pearson's correlation may be overly sensitive, as it penalizes any difference in the ordering of accounts regardless of the difference between their scores.

**6.3.2 Binary polarity prediction.** We report the results of a more lenient evaluation In Table 4, measuring the ratio of news accounts for which the polarity is correctly estimated. In this mode, the computed political orientation score is expected to be positive if the news source is considered to be conservative/Republican according to the reference poll, and vice versa (Eq 5). As shown in the table, the political orientation predicted using the social entity embeddings is accurate in almost all cases: prediction accuracy is 94% and 97% per the polls of 2014 and 2020, respectively. In each case, there was a single mistake in predicting the binary political bias. In error analysis, we found that the two faulty predictions apply to news accounts for which the number of contexts (i.e., followers) in our sample of Twitter was the lowest among

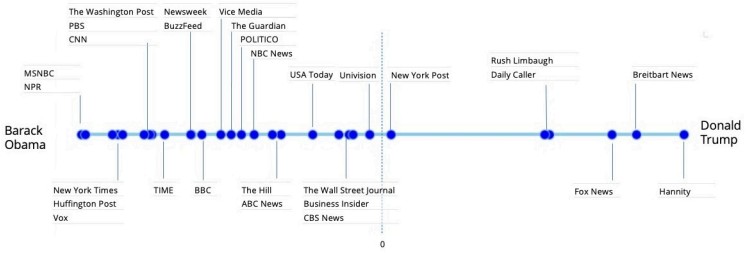

**Fig 3. Ranking of political polarity based on our embeddings.**

**Table 4. Results of assessing the political leaning of news sources as binary polarity.** Prediction accuracy is reported for all the accounts available per method ('all'), and for the news accounts that have embedding in all methods ('common').

| | Twitter accounts | SocialVec | Wikipedia2Vec | Wikidata RBG |
|---|---|---|---|---|
| Pew Research poll, 2014: | | | | |
| *All* | 31 | **0.94** (31) | 0.55 (28) | 0.32 (23) |
| *Common* | 22 | **0.95** (22) | 0.55 (22) | 0.27 (22) |
| Pew Research poll, 2020: | | | | |
| *All* | 30 | **0.97** (30) | 0.60 (28) | 0.27 (23) |
| *Common* | 22 | **0.95** (22) | 0.50 (22) | 0.23 (22) |

the accounts included in each poll, reinforcing the fact that sufficient context information is required for learning reliable low-dimension representations. In comparison, the Wikipedia2-Vec embeddings yield low accuracy scores of 55% and 60%, and the Wikidata graph embeddings yield poor accuracy of 32% and 27% per the two polls. As mentioned above, some of the news outlets were not included in Wikipedia of Wikidata. To account for differences in coverage, Table 4 also reports prediction accuracy for the subset of accounts that are represented by all methods ('common'). As shown, this evaluation mode shows the same trends.

Based on these results, we conclude that information about the political orientation of Twitter entities is well encoded in the social embedding space. In contrast, entity embeddings derived from factual sources fail to reflect political affinity. As reviewed in Sec. 2.3, related works devised specialized content analysis methods or collected and analyzed relevant network data ad-hoc for this purpose. Unlike those works, our approach is unsupervised and general in that we consider political bias as one aspect of social knowledge. Importantly, we believe that this approach may be employed for assessing additional types of social biases.

## 7 Case study II: Personal trait prediction

In our second case study, we utilize the SocialVec entity embeddings as features in inferring the personal traits of social media users. The task of personal trait prediction is of importance to personalization, recommendation, and Social Analytics applications, and has received substantial research attention (see Section 2.3). In our study, we project users onto the social embedding space based on the entities that they follow. We then use the resulting low-dimensional user representations to predict multiple socio-demographic traits of those users. In a set of supervised experiments, we show superior results using SocialVec compare with existing entity embeddings as predictive features for trait prediction, and further show advantage of our approach compared to content-based classification.

### 7.1 Dataset

We refer to a dataset of Twitter users labeled with respect to the following personal attributes: *age*, *gender*, *ethnicity*, *family status*, *education*, *income level*, and *political orientation* [48]. The user labels were obtained by means of crowd sourcing: human Workers were asked to make judgments about each user with regards to the specified properties based on public information in Twitter, including the user's self-authored description, their meta-data and the historical tweets posted by the user. In order to simply labeling, real-valued attributes were discretised into binary ranges. Originally, the dataset included 5,000 users, having subsets of the dataset labeled per category according to the availability of trait-related user information. We tracked 3,558 active user profiles overall in Twitter out of the original user set as of October

**Table 5. Personal trait prediction: Dataset statistics.**

| Attribute | Class Distribution | Profiles |
|---|---|---|
| Age | ≤25 y.o. (56%), >25 y.o. | 3,485 |
| Children | No (82%), Yes | 3,485 |
| Education | High School (67%), Degree | 3,485 |
| Ethnicity | Caucasian (57%), Afr. Amer. | 2,905 |
| Gender | Female (56%), Male | 3,475 |
| Income | ≤ 35K (64%), >35K | 3,485 |
| Political | Democrat (76%), Republican | 1,790 |

2020. This means that our experimental dataset is substantially smaller in size compared with the original dataset. Furthermore, since we retrieved user information some time after the annotation process was conducted, the category assignments of some attributes (for example, *age*) might have changed. Yet, since the labels are coarse-grained and were obtained based on subjective human judgement, we believe that labeling accuracy is not severely compromised. We note that beyond the labels being subjective, the dataset may present a selection bias, e.g., due to varying levels of self-exposure on social media by different sub-populations. Yet, we evaluate multiple alternative methods using the same data, and in the same conditions, where this forms a viable evaluation setup. Table 5 details the label annotation scheme, and the number of labeled users that comprise our dataset per category.

## 7.2 Methods

We perform supervised classification experiments, treating the prediction of the various attribute values as independent binary classification tasks. For each target attribute, we randomly split the set of labeled examples into distinct train (80%) and test (20%) sets, maintaining similar class proportions between these sets. Once the models are trained, prediction performance is evaluated against the gold labels of the test examples. Tuning was performed based on cross-validation performance using the training example sets.

**7.2.1 Entity embeddings as features.** We obtained information about the accounts followed by the users in the dataset using Twitter API, associating each user with the SocialVec embeddings of the entities which they follow. In learning, we follow the common practice of averaging the values of the bag-of-entity-embeddings that describe each user into a unified summary vector of the same dimension [60]. We then feed the averaged vector representation to a logistic regression classification network, in which the output layer consists of a single sigmoid unit. While we experimented also with multi-layer network architectures, we found this single-layer classifier to work best.

Table 6 includes detailed statistics about the number of entity embeddings that are available per user in the dataset using multiple entity embedding schemes. As shown, SocialVec provides a substantially larger coverage of the entities followed compared with the other methods. Specifically, the median number of encoded entity embeddings that are associated with each user is 104 using SociaVec versus 27 and 19 using Wikidata2Vec and Wikidata RBG, respectively. Consequently, the ratio of users that are poorly represented, being associated with less than 10 entity embeddings, is 4% using our method, vs. 23% and 33% using the alternative methods.

**7.2.2 Content-based trait prediction.** In our experiments, we consider textual content as an alternative information source for attribute prediction. Similar to previous works [48]), we

**Table 6. Personal trait prediction: Statistics of the number of popular account embeddings that are associated with each user in the dataset using the different entity embedding methods, and the proportions of users in the dataset that have a limited number of embeddings (less than 5 or 10) associated with them using each method.**

|  | Median | Average | Std.dev | <5 | <10 |
|---|---|---|---|---|---|
| SocialVec | 104 | 211.4 | 308.4 | 0.01 | 0.04 |
| Wikidata2Vec | 27 | 61.9 | 97.0 | 0.11 | 0.23 |
| Wikidata RBG | 19 | 43.1 | 69.8 | 0.20 | 0.33 |
| SocialVec ∩ Wiki. | 25 | 58.9 | 92.8 | 0.13 | 0.26 |

extracted up to 200 most recent tweets for each user in the dataset via Twitter API, obtaining $\sim 180$ tweets per user on average. The tweets authored by each user were converted into a 300-dimension text embedding vector using the pre-trained convolutional FastText neural model [61],which is a popular choice for representing the content of tweets in a low-dimensional form, e.g., [62]. Again, we averaged the resulting bag-of-tweet embeddings [63] to form a vector representation at user level, feeding the aggregated vector as input to a logistic regression network.

### 7.3 Results

Table 7 details classification performance per each of the target attributes in terms of the ROC AUC measure [49]. The table includes the results using SocialVec entity embedding features, as well as the results using Wikipedia2Vec and Wikidata RBG entity embeddings. As shown, SocialVec embeddings outperformed the other entity representation schemes by a large margin across all of the target attributes. For example, classification performance on *age* prediction is 0.74 in terms of ROC AUC using SocialVec versus 0.69 and 0.61 using Wikidata's and Wikipedia2Vec embeddings, respectively. On the target of *ethnicity*, ROC AUC using SocialVec embeddings is as high as 0.95 versus 0.86 and 0.70, and so forth.

In order to account for the gaps in coverage between the different embedding schemes (Table 6), we experimented with a restricted variant of SocialVec, where we discard the embeddings of entities that are not represented by any of the other methods from the user representation. As detailed in Table 6, this variant ('SocialVec∩Wiki') has similar coverage as the Wikipedia-based methods, with a median of 25 entity embeddings associated with individual users. As expected, classification performance using this limited feature set is lower, e.g., ROC AUC drops from 0.74 to 0.71 on the *age* category. Nevertheless, classification performance using the SocialVec embeddings remains superior on all of the target categories. Thus, we conclude that SocialVec entity embeddings are more informative compared with embeddings learned from factual knowledge bases on the social task of personal trait prediction.

**Table 7. Personal trait prediction results [ROC AUC].** *The results by Volkva *et al.* were obtained using a earlier version of the dataset which was substantially larger, and different tweets, and are therefore not directly comparable.

|  | Age | Children | Education | Ethnicity | Gender | Income | Political |
|---|---|---|---|---|---|---|---|
| SocialVec | **0.738** | **0.683** | 0.739 | **0.953** | **0.890** | 0.732 | **0.798** |
| Wikidata RBG | 0.686 | 0.614 | 0.690 | 0.864 | 0.803 | 0.682 | 0.694 |
| Wikipedia2Vec | 0.614 | 0.610 | 0.628 | 0.704 | 0.641 | 0.635 | 0.599 |
| SocialVec ∩ Wiki. | 0.705 | 0.665 | 0.698 | 0.924 | 0.859 | 0.709 | 0.748 |
| *Content-based*: |  |  |  |  |  |  |  |
| FastText | 0.695 | 0.575 | **0.740** | 0.785 | 0.768 | **0.748** | 0.654 |
| *Volkova et al.* [49] | *0.63* | *0.72* | *0.77* | *0.93* | *0.90* | *0.73* | - |

**Table 8. The top Twitter accounts that are characteristic to different subpopulations as measured using our datasets labeled with personal attributes and the Pointwise Mutual Information (PMI) measure.**

| Male | Female |
|---|---|
| *Ian Rapoport*, Sports writer and analyst (1.04) | *Chelsea DeBoer*, a reality TV persona (0.81) |
| *Chris Broussard*, Sports analyst, Fox Sports (1.02) | *womenshumor*, "tweets made for a woman" (0.80) |
| *Adam Schefter*, Sports analyst (1.02) | *Maci Bookout*, a reality television personality (0.76) |
| *Adrian Wojnarowski*, sports writer (1.01) | *Victoria's Secret*, a lingerie and beauty retailer (0.73) |
| *ESPNNBA*, The NBA games on ESPN (0.97) | *Country Words*, a country lyric page (0.71) |
| **White** | **Afro-American** |
| *starwars*, Star Wars on Twitter (0.80) | *KYLESISTER* (1.17) |
| *John Krasinski*, an actor, director and producer (0.78) | *Emmanuel Hudson*, actor (1.16) |
| *Luke Bryan*, a country music singer and songwriter (0.78) | *Erica Dixon*, TV personality (1.15) |
| *Country Words*, a country lyric page (0.77) | *Reginae Carter*, actress (1.15) |
| *Mark Hamill*, an actor and writer (0.75) | *Rasheeda*, a rapper (1.15) |
| **High-school** | **Academic** |
| *21 Savage*, a rapper, songwriter, and producer (0.45) | *The New Yorker*, an American magazine (1.30) |
| *AccessJui*, Music Producer (0.44) | *The Economist*, an international newspaper (1.20) |
| *Desi Banks*, a comedian, actor, and writer (0.42) | *Jack Tapper*, anchor and host at CNN (1.19) |
| *Lil Uzi Vert*, Rapper (0.41) | *The Wall Street Journal*, a business newspaper (1.19) |
| *Lil baby*, Rapper (0.40) | *Mashable*, a media and entertainment company (1.15) |
| **Republican** | **Democratic** |
| *Chick-fil-A*, a large fast food restaurant chain (1.15) | *Bryson Tiller*, a rapper (0.35) |
| *Carrie Underwood*, a Country singer (1.14) | *Kevin Gates*, a rapper (0.35) |
| *Tim Tebow* (1.13) | *Tami Roman*, a TV personality and rapper (0.34) |
| *BarstoolBigCat* (1.10) | *Iyanla Vanzant*, a TV personality (0.33) |
| *barstoolsports* (1.09) | *Lil Duval*, a stand-up comedian (0.33) |

**7.3.1 Data analysis.** To inspect the social information that is encapsulated in user-entity interactions, we examine the entity accounts which users tend to follow with distinctively different probability across class labels in the dataset. Table 8 shows the most distinctive accounts per label as computed using the pointwise mutual information (PMI) measure [64]. Concretely, for each entity account $e_i$ and target attribute $C$, the PMI between $e_i$ and attribute value $c_j$ is computed as $log \frac{Pr(e_i, c_j)}{Pr(e_i) \times Pr(c_j)}$, where $Pr(c_j)$ is the proportion of users in the dataset that are labeled with attribute value $c_j$, $Pr(e_i)$ is the proportion of users who follow entity $e_i$ regardless of the attribute label, and $Pr(e_i, c_j)$ is the joint probability of these events. In words, high PMI values indicate on distinctive as opposed to random entity-class co-occurrences.

Table 8 illustrates various social phenomena that are encoded in platforms like Twitter. We observe, for example, that the entity accounts which characterize male users specialize in sports, and that female users most distinctively follow accounts that belong to women. Likewise, the top accounts that characterise Afro-American users all belong to Afro-Americans (and vice versa). Similarly, young people (below 25 years of age) are associated with music bands and singers born in the 90s, while older users follow older TV hosts and celebrities. (Due to high class imbalance, the PMI values were lower for the *age* category, and this information was omitted from the table.) We further find that users with an academic degree distinctively tend to follow media accounts such as the New-Yorker and the Economist magazines, whereas non-academic users tend to follow rappers. Finally, the entity accounts that are most distinctive of Republican users in the dataset are non-political entities that are yet known as Republican oriented, including an account of country music, and the accounts of

Tim Tebow, a former professional football player, and the fast-food brand of Chick-fil-A, both of which are known for their conservative views; see "Tebowing" at https://en.wikipedia.org/wiki/Tim_Tebow, and https://en.wikipedia.org/wiki/Chick-fil-A: Same-sex marriage controversy. The accounts of the satirical Basrstool Sports are also identified as Republican oriented, echoing the findings of a recent poll; https://morningconsult.com/2020/07/24/barstool-sports-trump-interview-polling/. SocialVec entity embeddings encode these and other social patterns in an unsupervised fashion as observed from a large corpus.

**7.3.2 Comparison with other approaches.** We compare our trait prediction results with the more traditional yet popular approach of content-based classification. Table 7 presents our experimental results using the tweets authored by the users as evidence ('FastText'). The best results per trait are highlighted in boldface in the table. As shown, SocialVec entity modeling achieves top performance by a large margin on most (5/7) categories, where comparable results are obtained on the trait of *education*. (The difference in performance between FastText and SocialVec on the education category is not significant according to the McNemar $\chi^2$ statistical test.) Content-based classification achieves slightly better results in predicting the *income* level–0.75 versus 0.72 in ROC AUC. Notably, one's education and income levels are known to be manifested through their writing style [65].

Table 7 includes also the results previously reported by Volkova *et al.* [48, 49] per the original version of our reference dataset. They trained log-linear models using n-gram features extracted from the tweets posted by each user. Their results are not comparable to ours, as their dataset included many more labeled examples (see Sec. 7.1). Moreover, they relied on tweets obtained earlier in time. Nevertheless, despite the comparison being 'unfair' due to these limitations of our experiments, our results using SocialVec entity embeddings as features are comparable or exceed their results on the majority of the categories (4/6). More recently, researchers predicted *gender* and *ethnicity* using this dataset relying on the user's name as relevant evidence [66]. However, their AUC scores were 6-12% lower than those reported by Volkova and Bachrach [48], and are therefore obviously inferior to our results.

**7.3.3 Venues for further improvement.** There are several enhancements to our approach that can potentially improve prediction results further. First, the integration of network- and content-based evidence could improve prediction performance. In our experiments we have not observed an improvement when combining the two feature types, possibly due to the limited size of the dataset. Another common strategy is to enrich the users' representation based on their neighborhood in the social network, exploiting social homophily (assuming that the traits of the users are similar to the traits of their friends) [53]. Importantly, our approach is inductive, where we only learn entity embeddings once in an unsupervised fashion. This means that the accounts of any user, as well as the user's friends on the social network, may be represented using the available entity embeddings with no additional learning cost in a scalable fashion.

# 8 Discussion

We consider the learning of social entity embeddings as a first step towards the construction of a social body of knowledge. While researchers acknowledge the importance of social information modeling, this work is innovative in outlining a framework for eliciting general social knowledge at large scale. Below, we discuss the broad implications of this research, its limitations and future research directions. We also discuss ethical aspects involved in social knowledge modeling and its applications.

## 8.1 Implications of this research

We believe that this research can enhance applications that concern world knowledge in general and social knowledge in particular. Hereby, we discuss some potential research directions, placing emphasis on the inter-disciplinary field of Computational Social Science, and the prospects of personalized and socially contextualized text processing.

**8.1.1 Social knowledge exploration.** In this work, we inferred the political bias of news sources by simply computing cosine similarity between the embeddings of the specified news accounts and popular accounts of distinct political polarity. Characterising news sources and other organizations by their political slant at broader scale may assist in detecting and combating political biases and bubbles on social media. Likewise, one may track accounts which are socially similar to accounts marked with political extremism, so as to unveil accounts that spread harmful content, such as hate speech, conspiracy theories or fake news.More broadly, we believe that researchers may elicit various social insights by examining the affinity or polarity between entities of interest in the social embedding space. The automatic prediction of the users' socio-demographic traits is another direct benefit of our approach to social media analysis. This may allow, for example, to characterise the social groups that support social stances of interest [13]. The mapping of social media users onto the social embedding space presents an accurate and scalable method to obtain such informative.

**8.1.2 Social natural language processing.** Several researchers have recently claimed that natural language processing methods are limited in their capacity of decoding text meaning as long as they ignore social factors [10, 11, 67]. Ideally, the social and cultural background of the text author, or speaker, would be represented as meaningful context for correctly interpreting the text, or speech, generated by them. Otherwise, inferring the underlying intention from text or speech alone is prone to fail whenever the user opinion is conveyed implicitly, or when the text is sarcastic [68, 69]. The modeling of social factors is necessary also in applications of text generation, e.g., for the purpose of maintaining a socially consistent agent persona in dialogue management, or for generating culturally appropriate outputs by machine translation systems [10]. In a recent position paper, Flek [67] suggests that similar to contextual word embedding, neural models could be used to create large-scale social representations of users from online corpora that contain user metadata, so as to capture pretrained representations of user identities that encode their conversational styles, opinions, and interests. We believe that our approach for modeling social media users using vectors of social semantics forms an important step in this direction.

**8.1.3 Knowledge representation.** We found that the entity embeddings learned from social media also capture factual entity semantics (Table 2), having entities of the same semantic class and domain collocated in the embedding space. We have also shown that while many entities on social media are represented by knowledge bases like Wikipedia, the scope of entities that are included in SocialVec is broader and complementary to entities that emerge from factual sources. We therefore believe that SocialVec may be used as a valuable source of both social and factual world knowledge. Interestingly, in a set of preliminary experiments, we further found that SocialVec embeddings encapsulate relational arithmetics [70]. Similar to word analogies [16], SocialVec correctly predicts the missing entities in analogy queries such as {Android : Google :: Windows : ?}, and {DwightHoward : NBA :: DangeRussWilson : ?}, to be Microsoft and the National Football League (NFL). This suggest that SocialVec embeddings might support the automatic construction and completion of factual knowledge sources, serving as additional information source for inferring certain types of relational facts [4].

## 8.2 Limitations

There are naturally some limitations to our approach. SocialVec embeddings are learned from the social network of Twitter, which has biases; for example, Twitter users are younger and more Democrat than the general public; (https://www.pewresearch.org/internet/2019/04/24/sizing-up-twitter-users). In addition, while public figures like politicians and music artists typically maintain Twitter accounts, some entity types, e.g., locations, are not well-represented in this platform. Furthermore, accounts may be banned from social networks like Twitter. (A well-known example is the suspension of the private account of former president Donald Trump in January 2021). Another inherent limitation of embedding methods like SocialVec, regardless of the underlying information source, is that high-quality embeddings requires sufficient context statistics. Finally, the learned SocialVec entity embeddings capture social knowledge at some fixed point in time, whereas social networks are dynamic, and new entities emerge over time. We believe that the core of social knowledge changes slowly; consider, for example the political biases of new sources. Nevertheless, our approach of learning entity embeddings from a sample of the social network is efficient and can be readily applied to learn entity embeddings at different points in time. Repeated sampling of social knowledge may allow the study of temporal changes of social entity representations.

## 8.3 Ethical considerations

In our experiments, we use SocialVec entity embeddings as features in predicting the personal traits of individual users. In general, the task of learning user profiles from their digital footprints is well studied. In order to protect user privacy in applying such techniques in practice, users should be informed and approve the use of automatic personalization methods. We note that our approach, like other data-driven prediction method, may result in stereotypical user profiling. There are ongoing efforts within the research community that aim to mitigate and raise awareness to potential biases of this sort in using and interpreting machine predictions. A more detailed discussion of the ethical aspects involved in characterising users for the purpose of improving natural language processing is included in several recent position papers [10, 67].

In this research, we processed entity embeddings from public network information of sampled Twitter users. We discarded the users' identities, and processed the large-scale network information into a low-dimensional space. Thus, there are no traces of individual user information in the learned entity embeddings. The learned embeddings reflect general social contexts that characterise popular users, and do not pertain to any information that is associated with the entity accounts directly. We make SocialVec embeddings publicly available for research purposes, hoping to promote social knowledge modeling and exploration.

## 9 Conclusion

Researchers, practitioners and crowd workers have been constructing, maintaining and utilizing resources of world knowledge for decades. However, the existing knowledge sources that represent factual information fail to describe social aspects of knowledge. This work motivates and forms a first step towards the modeling of a general resource of social world knowledge.

We presented SocialVec, a framework for learning social entity embeddings from a large sample of a social network, and learned the embeddings of roughly 200,000 popular accounts using information about the accounts followed by 1.3 million users of Twitter. An exploration of the resulting embedding space showed that it encodes various social patterns and perceptions, as well as relational semantics. We demonstrated the applicability of SocialVec embeddings on two case studies of practical importance: inferring the political bias of media

accounts, and predicting the personal traits of social media users. Our evaluation on these socially-oriented tasks showed advantageous performance using the inferred social entity embeddings compared with existing entity embedding schemes which have been derived from information sources. Further, we have shown that that rather than devise ad-hoc methods to address each task, one may frame and process these and other tasks in terms of the learned social embedding space, in a general, simple and scalable fashion.

As next steps, we are interested in associating the entities that comprise the social knowledge base with semantic types, similar to factual knowledge bases which link entities with a semantic hierarchy [71]. We also wish to infer functional and social relationships between entity pairs or groups. A question of interest is whether and how can we detect and characterise phenomena of social polarity using quantitative measures within the social embeddings space. To name a few potential applications, we are interested in identifying and characterising social media accounts that spread hateful or uncivil content based on both content analysis and social affinity with toxic accounts in the social embedding space. We also wish to investigate correlations between public stances on social topics and socio-demographic factors and incorporate this information as context in stance prediction and sarcasm detection from text. In general, the social knowledge elicited in this work may enable the modeling of relevant social contexts in natural language processing applications, both generally and at individual-level.

## Author Contributions

**Conceptualization:** Einat Minkov.

**Data curation:** Nir Lotan.

**Investigation:** Nir Lotan, Einat Minkov.

**Methodology:** Nir Lotan, Einat Minkov.

**Software:** Nir Lotan.

**Supervision:** Einat Minkov.

**Writing – original draft:** Nir Lotan, Einat Minkov.

**Writing – review & editing:** Nir Lotan, Einat Minkov.

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
