## [Decision Letter · Decision Letter 0]

20 Dec 2022

PONE-D-22-18974Social World Knowledge: Modeling and ApplicationsPLOS ONE

Dear Dr. Minkov,

Thank you for submitting your manuscript to PLOS ONE. After careful consideration, we feel that it has merit but does not fully meet PLOS ONE’s publication criteria as it currently stands. Therefore, we invite you to submit a revised version of the manuscript that addresses the points raised during the review process.

We look forward to receiving your revised manuscript.

Kind regards,

Barbara Guidi

Academic Editor

PLOS ONE

Journal Requirements:

Reviewers' comments:

Reviewer's Responses to Questions

**Comments to the Author**

1. Is the manuscript technically sound, and do the data support the conclusions?

Reviewer #1: Yes

Reviewer #2: Yes

2. Has the statistical analysis been performed appropriately and rigorously? 

Reviewer #1: Yes

Reviewer #2: Yes

3. Have the authors made all data underlying the findings in their manuscript fully available?

Reviewer #1: No

Reviewer #2: Yes

4. Is the manuscript presented in an intelligible fashion and written in standard English?

Reviewer #1: Yes

Reviewer #2: Yes

5. Review Comments to the Author

Reviewer #1: The paper proposed an embedding method and used the co-follower network to learn social embeddings of entities. This method was used to perform two downstream tasks.

The method suggested and use cases are interesting and provide implications for other researchers in this area.

There are few items the author needs to address:

- Figures appear blank in the submission version

- Link to data/code is not provided.

- Author could mention the selection bias regarding the model predicting personality features of the users. It could be that demographics are correlated with related information users provide on Twitter e.g., younger users provide more info on their bio thus MTurk work could do a better job in judging their demographics.

- The paper is currently slightly too long particularly the abstract is wordy and hard to follow

Reviewer #2: The paper proposes an interesting study and it proposes SocialVec. The topic of the paper is really interesting, however it is not clear how the evaluation part has been organized. More detailed should be included, in particular the section related to the comparison. It si completely unclear the choices concerning the comparison. Please clarify this point in order to understand if this section is important or not, as it is.

6. PLOS authors have the option to publish the peer review history of their article (what does this mean?). If published, this will include your full peer review and any attached files.

Reviewer #1: No

Reviewer #2: **Yes: **Barbara Guidi

---

## [Author Response · Author response to Decision Letter 0]

2 Feb 2023

We thank the Editors and the reviewers for their constructive comments. Below are our detailed answers to each of the comments. 

REVIEWER 1:

There are few items the author needs to address:

Q1.1: Figures appear blank in the submission version

R: We believe to have followed the submission instructions, by which figures should be removed from the main file, and be submitted separately.

Q1.2: Link to data/code is not provided.

R: We declared that our data and code would be specified upon acceptance. Nevertheless, following this comment, we included these details in the revised manuscript. Specifically, the following text has been added to the Introduction section:

“… We make the SocialVec framework and the resulting entity embeddings as learned and applied in this work accessible to the research community, and believe that this has the potential of making a significant contribution to exploring social world knowledge as reflected in Twitter. Our code is available at github.com/nirlotan/SocialVecTraining. Another respository that contains the entity embeddings, as well as an API for assessing entity similarity, is available at https://github.com/nirlotan/SocialVec.”

Q: Author could mention the selection bias regarding the model predicting personality features of the users. It could be that demographics are correlated with related information users provide on Twitter e.g., younger users provide more info on their bio thus MTurk work could do a better job in judging their demographics.

R: We agree that there could be a selection bias in labeling the socio-demographic characteristics of users by crowd workers. Unfortunately, there is no alternative labeling method that is bias-free. For example, asking users to actively provide their details would only attract a subset of the population; or, labeling users based on their self-descriptions on Twitter would be inherently biased to those users who tend to share personal information online. That being said, we kindly note that the dataset used in our experiments, which was labeled by MTurk crowd workers, was collected and annotated prior to our work by other researchers (Volkova et al.). It is a public dataset, for which there exist previously published results using alternative methods, which we compare our results against. Nevertheless, to clarify the point raised, we added the following text to Section `6.1 Dataset’:

“We note that beyond the labels being subjective, the dataset may present a selection bias, e.g., due to varying levels of self-exposure on social media by different sub-populations. Yet, we evaluate multiple alternative methods using the same data, and in the same conditions, where this forms a viable evaluation setup.”

In addition, we discuss potential general biases involved in automatically predicting the personal traits of users in Sections `7.2 Limitations’ and `7.3 Ethical considerations’. 

Q: The paper is currently slightly too long particularly the abstract is wordy and hard to follow.

R: Thank you for this comment. We have made a substantial effort to make the whole paper less wordy and more fluent. Overall, this effort resulted in a reduction of the paper length by more than 10%. In particular, we have edited the abstract, having it shortened as a result by roughly 25% (from 368 to 283 words). A document that highlights the tracked changes is submitted as part of this revision, illustrating the many edits performed to the revised manuscript.

REVIEWER 2:

The paper proposes an interesting study and it proposes SocialVec. The topic of the paper is really interesting, however it is not clear how the evaluation part has been organized. More details should be included, in particular the section related to the comparison. It is completely unclear the choices concerning the comparison. Please clarify this point in order to understand if this section is important or not, as it is.

R: Thank you for noting this issue. To clarify the setup of the evaluation, and the underlying motivations for this setup, we have created a new main section in the revised paper (Sec.4) which is titled `4. Evaluation’. 

We then added Section `4.1. Evaluation methodology’, which includes two main paragraphs. The first paragraph discusses `intrinsic vs. extrinsic evaluation’. Below are some excerpts from this paragraph:

“Unfortunately, there are no relevant human-judged benchmarks that assess entity similarity. Instead, we take a direct look at inter-entity similarities by exploring the entities that are most similar to example entities of interest in the learned social embedding space. …. This intrinsic evaluation is presented and discussed in Section 4.2. Nevertheless, we mainly place our focus on extrinsic evaluation, where we gauge the utility of the learned social entity embeddings for end applications which involve social inference. First, we assess the political bias of news sources in terms of entity similarity in the social embedding space. Second, we predict the personal traits of individual Twitter users based on the social embeddings of entities that they follow. These studies are presented in Sections 5 and 6, respectively.”

The second paragraph in Section 4.1 discusses our comparison against alternative methods and baselines. Here is an excerpt from this paragraph:

“In evaluating SocialVec on end tasks, we review and compare our results against alternative methods, which have been designed and applied per those target tasks and experimental datasets. In addition, we contrast our social entity embeddings with existing entity embedding schemes, namely, Wikipedia2Vec and Wikidata RBG embeddings. … To the extent that SocialVec achieves higher performance on tasks of social inference, this implies that it is superior in capturing dimensions of social meaning.”

We believe that this section improves the understanding of how we apply and evaluate the proposed framework.

---

## [Decision Letter · Decision Letter 1]

14 Mar 2023

Social World Knowledge: Modeling and Applications

PONE-D-22-18974R1

Dear Dr. Minkov,

We’re pleased to inform you that your manuscript has been judged scientifically suitable for publication and will be formally accepted for publication once it meets all outstanding technical requirements.

Kind regards,

Barbara Guidi

Academic Editor

PLOS ONE

Additional Editor Comments (optional):

Reviewers' comments:

Reviewer's Responses to Questions

**Comments to the Author**

1. If the authors have adequately addressed your comments raised in a previous round of review and you feel that this manuscript is now acceptable for publication, you may indicate that here to bypass the “Comments to the Author” section, enter your conflict of interest statement in the “Confidential to Editor” section, and submit your "Accept" recommendation.

Reviewer #1: All comments have been addressed

Reviewer #2: All comments have been addressed

2. Is the manuscript technically sound, and do the data support the conclusions?

Reviewer #1: Yes

Reviewer #2: Yes

3. Has the statistical analysis been performed appropriately and rigorously? 

Reviewer #1: Yes

Reviewer #2: Yes

4. Have the authors made all data underlying the findings in their manuscript fully available?

Reviewer #1: Yes

Reviewer #2: Yes

5. Is the manuscript presented in an intelligible fashion and written in standard English?

Reviewer #1: Yes

Reviewer #2: Yes

6. Review Comments to the Author

Reviewer #1: The author has address all of my comments and I'm happy with the current state of the manuscript. I do not have further comments.

Reviewer #2: The quality of the manuscript has been improved and all the previous requests have been addressed. For this reason, the paper could be accepted as it is.

7. PLOS authors have the option to publish the peer review history of their article (what does this mean?). If published, this will include your full peer review and any attached files.

Reviewer #1: No

Reviewer #2: **Yes: **Barbara Guidi

---

## [Editor Report · Acceptance letter]

20 Mar 2023

PONE-D-22-18974R1 

Social World Knowledge: Modeling and Applications 

Dear Dr. Minkov:

I'm pleased to inform you that your manuscript has been deemed suitable for publication in PLOS ONE. Congratulations! Your manuscript is now with our production department. 

Kind regards, 

on behalf of

Dr. Barbara Guidi 

Academic Editor

PLOS ONE